# Habitat Characteristics Coincidence of Dead and Living Long-Tailed Gorals (*Naemorhedus caudatus*) According to Extreme Snowfall

**DOI:** 10.3390/ani11040997

**Published:** 2021-04-02

**Authors:** Hee-Bok Park, Sungwon Hong

**Affiliations:** 1Restoration Research Team (Mammals), Research Center for Endangered Species, National Institute of Ecology, Yeongyang 36531, Korea; marinecat80@nie.re.kr; 2Department of Animal Science and Biotechnology, Kyungpook National University, Sangju 37224, Korea; 3Department of Horse, Companion, and Wild Animal Science, Kyungpook National University, Sangju 37224, Korea

**Keywords:** survival, climate change, population viability analysis, ensemble species distribution model

## Abstract

**Simple Summary:**

The long-tailed goral (*Naemorhedus caudatus* Milne-Edwards) is a critically endangered herbivore in South Korea. From March to June in 2010, 24 animals were found to have died due to heavy snowfall in the Wangpi Stream basin. In this study, we hypothesized that gorals that died due to snowfall are low-status individuals that lived in the sub-optimal or non-suitable areas. The results suggested that the sites where dead gorals were found were highly related to typical goral habitats. The optimal goral habitats could become uninhabitable following heavy snowfall. Most of the dead animals were pregnant females or were young, implying that they could not escape their primary habitats due to lower mobility. Thus, when there is a climate catastrophe, the optimal goral habitats should be considered for rescue and artificial feeding.

**Abstract:**

The long-tailed goral (*Naemorhedus caudatus*) is a critically endangered herbivore in South Korea. Despite government efforts to recover the population through reintroduction programs, the animal remains vulnerable to heavy snowfall. From March to June 2010, 24 animals were found dead due to heavy snowfall in the Wangpi Stream basin. In this study, we hypothesized that gorals that died due to snowfall are low-status individuals that lived in the sub-optimal or non-suitable areas. Using the occurrence data from extensive field surveys from 2008 to 2010 in the Wangpi Stream and the carcass location data, we (1) defined the goral habitat characteristics and (2) compared the habitat characteristics between dead and living gorals using ensemble species distribution modeling. The results suggested that the sites where dead gorals were found were highly related to typical goral habitats. These results implied that the optimal goral habitats could become uninhabitable following heavy snowfall. Most of the dead animals were pregnant females or were young, implying that they could not escape their primary habitats due to lower mobility. Thus, when there is a climate catastrophe, the optimal goral habitats should be considered for rescue and artificial feeding.

## 1. Introduction

Extreme winter seasons can affect the foraging behaviors of herbivores and can result in increased fatality [1]. Food deprivation causes a deficiency in energy in herbivores [2]. Moreover, low temperatures during winter lead to high-energy requirements [3]. Besides this, warm spells in the winter with rainfall (rain-on-snow) can occasionally cause “icing,” which restricts access to food, resulting in starvation and lower survival and fecundity rates [4].

Long-tailed goral (*Naemorhedus caudatus*) belongs to four *Naemorhedus* species of Caprinae, which are endangered and are one of the least studied montane bovid in the world. The animal is distributed in Eastern and Northern Asian mountains, including Russia, China, and Korea [5,6]. This species is listed as a vulnerable species by the International Union for Conservation of Nature and Natural Resources (IUCN) [7]. Whereas, in South Korea, the species has been protected as a critically endangered species by the Ministry of Environment and natural monument no. 217 by the Cultural Heritage Administration. As of 2018, the estimated population was only 679 individuals in representative habitats [8]. This small population has survived as they inhabit deep valleys and cliffs where humans and predators cannot easily approach them [9].

The species has been protected under the reintroduction policy of the Korean government. However, they still face several threats, such as road-kills and climate change [10,11]. In particular, heavy snowfall represents a mortality risk [12]. Gorals evolved to have shorter forelegs than hind-legs in order to quickly climb cliffs; however, this means that, when they are trapped by heavy snow, they cannot escape and are likely to die from starvation [13].

Two notable incidents occurred due to abnormal heavy snowfalls in South Korea. The first was, from March 1964 to February 1965, approximately 6000 gorals were trapped by snow and died. The second was, from March to June 2010, 24 gorals were found dead in Uljin, a small countryside county [12,14]. In 2010, the fresh snow cover depth was the second highest for 10 years, and the volumes of sunshine and air temperature for six months (from January to June) were remarkably lower than those between 1971 and 2000 (Figure 1). In recent years, strict policies reduced their fatality from human activities, resulting in severe climatic conditions being the most crucial factor decreasing the goral population currently. Thus, it is vital to understand why the gorals die due to abnormal climatic conditions to reduce the fatality of future catastrophes.

Gorals live in flexible, hierarchical social groups [13,15]. They follow a polygyny mating system and live in groups that can comprise over 11 animals, depending on resource availability; however, the males usually live in solitude, except during the rutting season [16,17]. Gorals prefer to live on exposed rocks or on the south-facing regions of steep valleys where snow melts easier and can access their preferred food. However, gorals that are lower in the social hierarchy or yearlings could be expelled to live in areas more vulnerable to heavy snowfall [18]. Besides, in areas with soil with a weak draining capacity, snow does not melt quickly, which can be fatal for gorals. Thus, we hypothesized that the characteristics of both the preferred habitats of gorals and the locations where dead gorals were found would be different. In this study, we aimed to (i) identify the habitat characteristics of gorals, (ii) compare habitat characteristics, and (iii) predict the vulnerable areas.

## 2. Materials and Methods

### 2.1. Study Area

Extensive field surveys were conducted in the Wangpi River basin in the eastern part of South Korea (Figure 2a). The selected area was the remote not-industrialized countryside (urban area percentage: 1.00%) [17], which had a high forest cover (over 92.17%) and many streams. A small portion of the land is used for agriculture (approximately 4.96%). In 2018, the Korean red pine (*Pinus densiflora* Siebold and Zucc.; Pinaceae) was the dominant flora (approximately 46%), while Oak (Fagaceae spp.) occupied 37.66%. Over 90% of trees in the area are aged between 30 and 60 years (Korean National Spatial Data Infrastructure Portal, http://www.nsdi.go.kr (accessed on 1 April 2021).

The area has one of the highest goral populations in South Korea [14], and gorals in the area have distinct genotypes from other populations, probably due to the natural climatic and geographic barriers, such as a high mountainous plateau [19]. However, the study area does contain areas with different population densities. The northern areas (red polygon in Figure 2b of the Gwangcheon Stream watershed have higher population densities than the southern areas (violet polygon of Figure 1b, evidenced by the density of the traces such as feces, marking, etc. [20].

### 2.2. Field Survey and Carcass Survey

Surveys were conducted based on the line transect method. The survey sites were determined based on remoteness from urban areas and the proportion of the rocky regions by reviewing the literature on goral habitats. Most of the sites were steep rocky valleys with a small human population [13]. The surveys were conducted five times (March, April, May, June, and September) in 2008, three times in 2009, and twice in 2010.

To elucidate the habitat of the gorals, we searched for hairs of the gorals indicating their preferred habitats. When gorals spend more time in a given area or mark their territory by using their horns, they are likely to release hairs at the sites or have hair caught in branches [9]. We used hair and feces to determine the preferred habitats of the gorals, as footprints can be confused with those of other herbivores such as Siberian roe deer (*Capreolus pygargus* Pallas) and feral goats (*Capra hircus* Linnaeus) [9,20]. The location of all hair samples was recorded using a GPS recorder (Garmin, 60csx). The forest rangers and residents found the carcasses of dead gorals, and they were only found in the northern parts of the study area [21].

### 2.3. Environmental Factors Influencing the Habitats of Goral

Based on previous studies, we tentatively selected 27 environmental factors to characterize the habitat conditions of the dead and living gorals (Table 1). Previous studies only considered a few landscape factors, such as altitude, slope, aspect, and anthropogenic factors, such as distance to the road and trail. However, in this study, we considered more factors, such as distance to agricultural fields, factories, human residences, and soil characteristics to explore the primary goral habitats (Table 1) [14,22,23,24]. We considered the density of the anthropogenic, landscape, and forest variables within the male home range (1.46 km^2^; radius: 682 m measured by GPS collars) [25]. We primarily considered the male home range because typically, the male home range is larger than that of the female due to their solitariness, as such, their range would often encompass the female home range. As soil characteristics might affect the mortality or habitat of the gorals due to the differing snow melting ability related to the thermal cover or vegetation distribution, we added these characteristics as important variables [26]. The study sites are relatively small fragmented habitats, and we did not consider the meteorological factors to differentiate the habitats. The resolution of all variables was resampled to 30 x 30 m using the resample function of the raster R package [27].

### 2.4. Definition of Habitat Characteristics of Gorals

We built different species distribution models (SDMs) to define the habitat suitability and relationship between goral occurrence and environmental factors [30,31]. We used goral presence data from the line transect method so that the data could be spatially autocorrelated. Thus, before building the models, we cleaned some occurrence points within the home range of goral using the spThin package [32]. We adapted eight model algorithms [33]: two envelopes and distance-based approaches (BIOCLIM and Domain [34]), two regression-based approaches (generalized linear models (GLMs [34]) and generalized additive models (GAMs [35]), and four machine-learning methods (classification and regression trees (CART [36]), random forests (RFs [37]), boosted regression trees (BRTs [38]), and Maxent [39]). The Fine-tuning of each model was conducted according to the instructions from the practices by Dr. Zurell (https://www.damariszurell.io/HU-GCIB/5_SDM_algorithms.html) (accessed on 1 April 2021).

Before building the model, we evaluated the collinearity and removed the collinear variables by assessing the variance inflation factors (cut-off value = 5) with the vifstep function of the “usdm” packages in R Studio v1.1.456 [40]. Then, we excluded eight variables, especially the distance to a rice paddy, soil drainage, tree age class, degree of rock exposure, distance to a factory, stream density, aspect, and tree diameter class. We explored the relationships between the goral presence and each environmental variable in order to identify the relevant variables. First, we validated the predictive power by adding the variables using 10-fold cross-validation procedures to estimate the area under the ROC curve (AUC ≥ 0.80) and True Skill Statistic (TSS) (≥0.6). The procedure was repeated 50 times, and the results were averaged [41,42]. If the added variables did not improve the model accuracy (i.e., average variable importance ≤ 5), we extracted them and added other ones. Second, if the plotted models did not show reasonable relationships, we did not consider those variables [43]. For example, when the relationship between the presence of gorals and the distance to streams exhibited quadratic shapes, which could not be explained, we considered the corresponding variable as a non-influential factor. Finally, after determining all the influential variables, we conducted 10-fold cross-validation and selected the model algorithms that showed a high goodness of fit (ROC curve ≥ 0.80 and TSS ≥ 0.60). We then used the median of the model performance [30]. We used the varImp function of the caret R package [44] to define the variable importance. For the algorithms for which the R packages do not provide importance (BIOCLIM, Domain, and Maxent from maxnet R package), we calculated the importance based on the ROC curve (importance = 1 − AUC) by excluding each included variable from the full combination of variables [44,45]. Then, the relative importance was calculated by dividing the variable importance by the total importance.

### 2.5. Definition of Site Characteristics of Dead Gorals

When we determined the goral presence sites, we clipped the areas in the upper study regions (red polygon in Figure 2b due to considerably different goral densities [20]. Then, we made 50 background points in the polygon to differentiate the habitat characteristics of dead gorals using the randomPoints function of the dismo R package [27]. By following the previous procedures (identification of relevant variables) of variable selection using ensemble SDMs, we defined the habitat characteristics of dead gorals (Figure 3).

## 3. Results

### 3.1. Habitat Characteristics of Gorals in Uljin

The models that reached fair levels of goodness-of-fit were RFs (AUC = 1.00 ± 0.00 and TSS = 1.00 ± 0.00; Average ± Standard error) and BRTs (AUC = 0.81 ± 0.002 and TSS = 0.64 ± 0.003; Appendix A). The ensemble model had high levels of goodness-of-fit (AUC = 0.87 and TSS = 0.69) and suggested that the distance to the field (m), coniferous tree density (m^2^), and distance to the trail (m) influenced the goral habitats (Figure 4). Distance to the field (m, 0.59 ± 0.09) was identified as the most influential factor (Figure 4a). The probability of the goral increased in regions with a high density of coniferous trees and increased as the distance to fields and trails increased (Figure 4b).

It was observed that there were large clusters located in the central longitudinal areas (Figure 5), whereas there were a few optimal areas in the western regions. However, the suitability was distinctively lower in eastern areas near the seashore (Figure 5a). When thresholds were divided into presence and absence areas, the clusters in longitudinal centers were fragmented and were weakly connected in some areas (dotted circles in Figure 4b). The western regions were remarkably fragmented (Figure 5b).

### 3.2. Site Characteristics of Dead Gorals

The models that had fair levels of goodness-of-fit were RFs (AUC = 1.00 ± 0.00 and TSS = 1.00 ± 0.00), Maxent (AUC = 0.81 ± 0.003 and TSS = 0.65 ± 0.01), and Domain (AUC = 0.80 ± 0.002 and TSS = 0.64 ± 0.004; Appendix A). The ensemble model scored high levels of goodness-of-fit (AUC = 0.88 and TSS = 0.65) and suggested that the influential factors were similar to those of the preferred habitats of the goral, such as distance to the field (m) and coniferous tree density (m^2^). However, aspect (°) was newly selected (Figure 6). The distance to the field (0.41 ±  0.03%) was identified as the most influential factor. Other factors were also severely influential (Figure 6a). The probability of goral mortality due to snowfall generally increased with an increase in distance to fields and areas with higher coniferous tree densities, as found in south-east facing areas (Figure 6b).

A large cluster of areas with a high goral mortality rate is located in the central longitudinal regions (Figure 7a). Within the cluster, some areas were fragmented because the map excluded the south-east facing areas (Figure 7b).

## 4. Discussion

The long-tailed goral is a critically endangered species and is one of the least studied species of montane ungulates [6,7]. As defining the distribution and habitat characteristics was conducted prior the study to make conservation plans for the endangered species, investigating the ecology of the habitat of gorals is urgently needed [46]. Several studies explored the habitat characteristics of gorals. Most of them focused on landscape factors, such as slope, altitude, aspect, and distance to streams [14,22,23,25]. However, as gorals can be flexible regarding these factors, the results of these studies cannot be used to generalize ideal habitat characteristics. For example, individual herbivores of the same species, including gorals, can adapt to different altitudes based on food abundance and human disturbances [47]. Therefore, the results of previous studies on goral habitats did not have a complete view [13,14]. The results of our habitat analysis suggest that human disturbance was the primary factor influencing the goral habitat.

By exploring the relevant variables of the 27 tentative environmental factors, the habitat characteristics of gorals suggest that the species could be critically susceptible to human activity (Figure 4 and Figure 5) [24]. Large distances from fields and trails and the high density of coniferous trees indicate habitats that are less disturbed by humans and are more remote [13]. Areas with a high density of broadleaf trees provide optimal living conditions for gorals. However, these areas also have a high human population [48]. Thus, in our study, we found that gorals preferred to live in areas with a high density of coniferous trees, even though areas with broadleaf trees offered continuous food availability [49]. It is difficult for gorals to avoid human activity as activities often occur in goral habitats, as the areas of optimal habitat for gorals are shrinking and fragmented in terms of the edge effect. Therefore, it is crucial to review the ongoing construction and land use changes within the habitats of the gorals. In Uljin, as highways were constructed, the government planned for local roads to be restored as non-paved roads. While the restoration could provide more suitable habitats, the construction and restoration impacts should be thoroughly investigated initially. The cable car construction in the Seorak national mountain, which has the highest goral population, should be reconsidered.

Habitat selection by gorals could be dependent on their rank within the hierarchy [50]. Gorals of low-status may suffer from reduced access to resources, whereas those in the higher-order generally have priority access [51]. Within the group, the location of territory and its suitability can also be determined by rank [52]. For example, the optimal habitats to survive heavy snowfall could be occupied by the individual with the highest ranking. Thus, analyzing the spatial positions related to the ranks could suggest some reasons why gorals have been found dead in abnormal climates during the winter season.

The difference between the preferred habitats of gorals and high mortality areas was rejected, which disagrees with our hypothesis. We found that the characteristics of areas with a high mortality rate also represented the preferred habitats of gorals. Variables such as large distances to the field and high density of coniferous trees are the same variables that explain the preferred habitats of gorals. For example, south-east facing areas are the preferred habitats of gorals because snow melts quicker due to more sunlight exposure, as the preferred diet grows more easily on these aspects [15,49]. However, our results suggested that a high mortality rate also characterized these areas. Thus, we concluded that mobility might be the most critical factor affecting the survival of gorals.

Most of the dead gorals (89.4%) were yearlings or pregnant females (Table 2) [21]. The autopsy report from the dead gorals suggested that they were starved but had no diseases. Generally, their mobility is less than that of males and nongravid females [53,54]. In low temperatures, pregnant females spend more energy maintaining their body temperature for their cubs [55], whereas yearlings usually have less physical ability than adults. When heavy snowfall occurs, most gorals tend to migrate to lower altitudes [12,25,56]. However, if gorals are unable to escape from stacked snow, their home ranges become smaller, and they are forced to stay in solitude in their preferred areas, such as rocky south-facing areas [15,25,57]. While these areas offer food, as the weather becomes extreme, the vegetation cover decreases. The lower food availability causes gorals to become exhausted and can potentially starve.

If the “age” and the “dominant rank” of animals are proportional, intraspecific competition (i.e., male > female and adult > juvenile) might also affect mortality. However, our previous research showed that the age and dominant rank of the gorals were not proportional [17]. The number of adults (5 ± 3) was higher than those of juveniles (3 ± 2). Besides, the number of females (4.5 ± 0.5) was higher than males (2.5 ± 0.5). However, even though the sites were the representative habitats for the goral, the number of sites in the study area was only two. While the information was limited, any study reporting that intraspecific competition of gorals can cause high mortality for females and juveniles is unknown to us.

The survival of pregnant females and yearlings is crucial to increase the goral population [10]. The remote countryside offers gorals an optimal habitat (Figure 4). However, when extreme snowfall occurs, these areas become the regions with the highest mortality rates (Figure 5). As most gorals migrate to lower altitudes due to heavy snowfalls, it was believed that the priority rescue areas should be at lower altitudes. However, our results suggest that high altitude remote areas with high goral population density should be focused on, as rescuing gorals belonging to vital age classes is necessary to increase their population. Thus, the primary sites for rescue and artificial feeding should be considered and established at optimal goral habitats in a climate catastrophe event.

## 5. Conclusions

Extreme winter seasons can affect the foraging behaviors of herbivores and result in an increased fatality. This study hypothesized that gorals that die due to abnormal weather conditions could be low-status individuals who live in sub-optimal or non-suitable areas. The habitat comparisons between dead and living long-tailed gorals using SDMs suggested that the sites where carcasses were identified were highly related to typical goral habitats, thus contradictory to our hypothesis. The results implied that optimal goral habitats could become deadly following heavy snowfall. The dead animals were mainly pregnant females or young individuals who could not escape their primary habitats due to their low mobility. Thus, we concluded that primary sites for rescue and artificial feeding should be established in the optimal goral habitats when there is a climate catastrophe.

## Figures and Tables

**Figure 1 animals-11-00997-f001:**
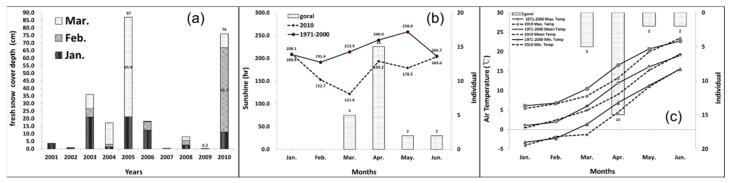
Climatic conditions in 2010 compared to those in other years. (**a**) Monthly average fresh snow cover depth (cm) for three months (January: dark bar, February: intermediate dark bar, and March: light bar) from 2001 to 2010, (**b**) monthly average sunshine (hr) and (**c**) air temperature for six months (from January to June) of 2010 (dotted line), compared to that between 1971 and 2000 (line) and the monthly number of dead gorals.

**Figure 2 animals-11-00997-f002:**
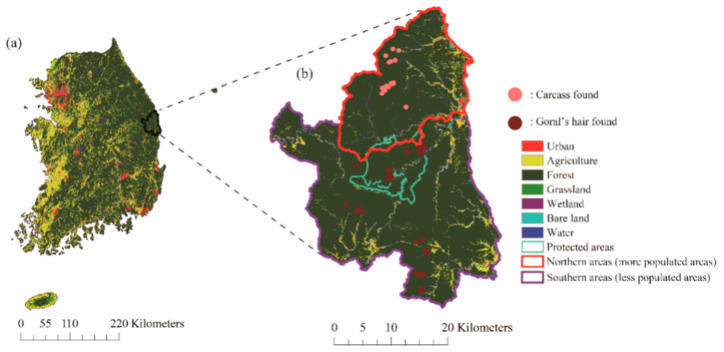
(**a**) Land use in South Korea and (**b**) study sites (red: urban, yellow: agriculture, dark green: forest, light green: grassland, violet: wetland, emerald: bare land, and blue: water). The emerald polygon represents the protected area. Light and dark red points represent the locations of goral carcasses and hair samples, respectively.

**Figure 3 animals-11-00997-f003:**
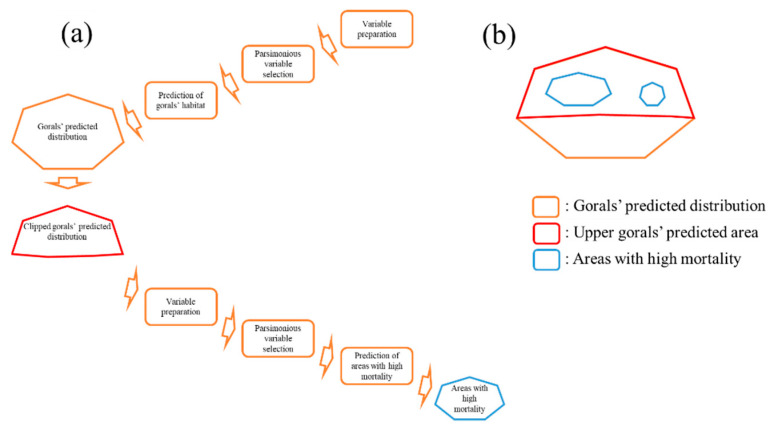
(**a**) Procedures of ensemble species distribution modeling to predict the potential distribution of gorals (orange polygon) and areas with a high mortality rate (blue polygons). (**b**) An example of the predicted habitat of the gorals and areas with a high mortality rate.

**Figure 4 animals-11-00997-f004:**
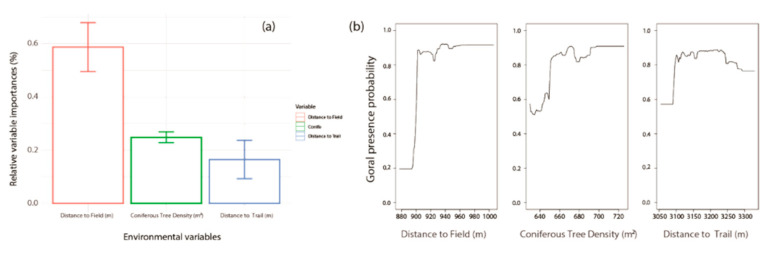
Three selected variables of ensemble model with the algorithms of Random Forest and Boosted Regression Tree: (**a**) relative variable importance (%) and (**b**) relationship between goral presence probabilities and the three relevant factors.

**Figure 5 animals-11-00997-f005:**
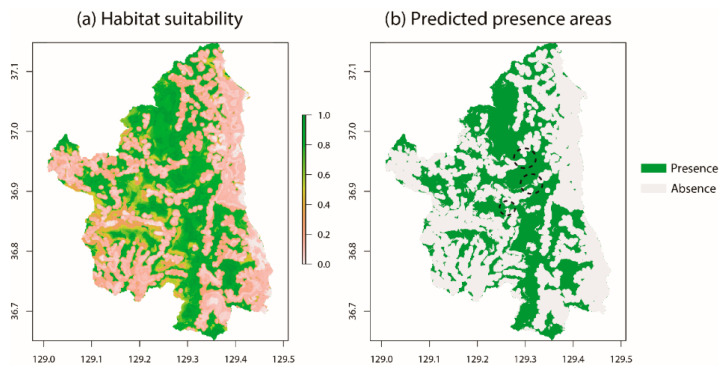
Habitat suitability (**a**) and predicted presence-absence areas (**b**) within the study sites. (**a**) Areas with high suitability are represented in green, whereas low suitability areas are represented in pink. (**b**) Predicted presence areas are represented in green, whereas absence areas are represented in light gray. The dotted circles indicate areas where the connectivity between fragmented areas is low.

**Figure 6 animals-11-00997-f006:**
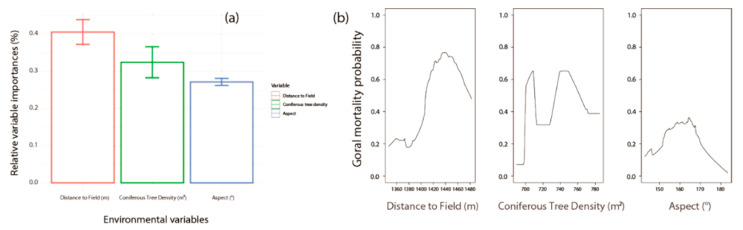
Three selected variables of the ensemble model with the algorithms of Random Forest, Maxent, and Domain. (**a**) Relative variable importance (%) and (**b**) relationship between goral presence probabilities and the three relevant factors.

**Figure 7 animals-11-00997-f007:**
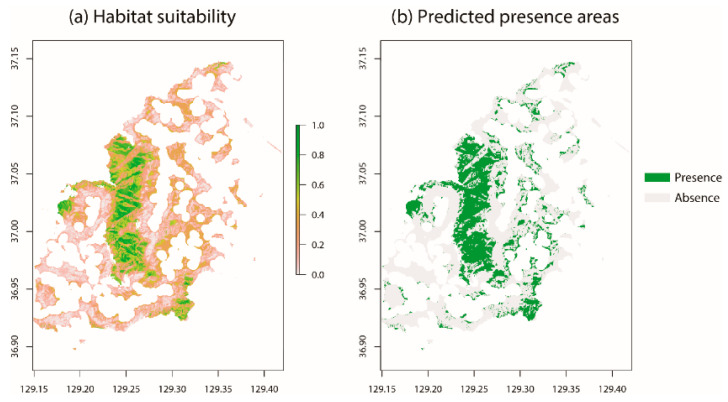
Mortality probability due to heavy snowfall (**a**) and predicted mortality areas (**b**) within the study sites. (**a**) Areas with high mortality are represented in green, whereas low mortality areas are represented in pink; (**b**) areas of predicted death are represented in green.

**Table 1 animals-11-00997-t001:** Description, abbreviation, related references, and source map (website, resolution, and making year) of the 27 selected variables.

No	Classification	Factors	Description and Hypothesis from Literature	Reference	Source Map
1	Anthropo genic fac tors	Distance to factory (m)	Human activity can negatively affect the goral habitats.	[22,23,24,28]	EGIS (https://egis.me.go.kr (accessed on 1 April 2021), 30 × 30 m, 2007)
2		Distance to road (m)			
3		Road density (m^2^/1.46 km^2^)			
		Human population density (n/1.46 km^2^)			
4		Distance to rice paddy (m)			
5		Distance to field (m)			
6		Field density (m^2^/1.46 km^2^)			
7		Distance to trail (m)			
8	Forest	Distance to deciduous forest (m)	Types of forest can influence goral’s feeding and marking sites.	[13,15,23]	National Spatial Data Infrastructure Portal (http://nsdi.go.kr (accessed on 1 April 2021), 5 × 5 m, 2018)
9		Density of deciduous forest (m^2^/1.46 km^2^)			
10		Density of coniferous forest (m^2^/1.46 km^2^)			
11		Density of forest (m^2^/1.46 km^2^)			
12		Age class (1 (young) to 9 (old))			
13		Diameter class (0 (young) to 3 (large))			
14		Crown density (1 (sparse) to 3 (dense))			
15	Landscape	Distance to stream (m)	Gorals used to live near the waterway.	[14,15]	EGIS (https://egis.me.go.kr (accessed on 1 April 2021), 10 × 10 m, 2013)
16		Stream density (m^2^/1.46 km^2^)			
17		Lake density (m^2^/1.46 km^2^)			
18		Altitude (m)	Gorals used to live at high altitudes to avoid anthropogenic disturbance.	[17,24]	Biz-gis (www.biz-gis.com (accessed on 1 April 2021), 30 × 30 m)
19		Aspect (1 to 360)	As snow can be more easily melt, the gorals prefer the habitats in southern regions.	[14,15,29]	Biz-gis (www.biz-gis.com (accessed on 1 April 2021), 30 × 30 m)
20		Slope (°)	Gorals prefer to live in the deep valleys to avoid humans.	[22,28]	Biz-gis (www.biz-gis.com (accessed on 1 April 2021), 30 × 30 m)
21	Soil map	Soil drainage (1 (bad) to 4 (very good))	If water cannot be displaced, snow becomes stacked. The large quantity of snow can negatively affect the goral’s escape from the snow.	[14]	National Spatial Data Infrastructure Portal (http://nsdi.go.kr (accessed on 1 April 2021), 25 × 25 m, 2018)
22		Degree of rock exposure (1 (little) to 4 (a lot))	Gorals prefer to live in front of large rocks to escape from predation. Besides, stacked snow melts more quickly than in other soil.	[13,15,26]	
23		Soil type (1 (no) to 3 (many))	Strongly eroded soil stacked by snowfall could make gorals fall easily.		
24		Degree of wind exposure (1 (exposed) to 3 (closed))	Exposure to wind can cause snow to remain for a long period.	[4,14]	
25		Distance to wet soil (m)	Moist soil delays the rate of snow melting.	[14]	
26		Wet soil density (m^2^/1.46 km^2^)			
27		Slant type (1 (rising slope) to 3 (falling slope))	Gorals prefer to live in convex slopes to escape from predation.	[13]	

**Table 2 animals-11-00997-t002:** Carcass information (observed and collected day, sex, age class (yearling: age < 1, juvenile 1 < age < 3, adult > 3), estimated ages, and pregnancy; CGRB, 2010).

No	Observed Day	Collected Day	Sex	Age Class	Estimated Ages	Pregnant
1	2010.03.16	2010.03.19	M	Juvenile	>1	
2	03.23	04.01	F	Adult	>15	Y
3	03.25	03.26	M	Yearling	<1	
4	03.28	04.01		Adult	5–6	
5	03.31	04.02		Juvenile	>1	
6	04.07	04.07	F	Adult	>10	Y
7	04.03	04.08	F	Adult	>10	Y
8	04.09	04.09	M	Juvenile	>1	
9	04.09	04.09		Yearling	<1	
10	04.10	04.10		Juvenile	<1	
11	04.10	04.10	M	Juvenile	>1	
12	04.11	04.11	F	Adult	5–8	Y
13	04.10	04.10	M	Juvenile	<1	
14	04.11	04.11	F	Juvenile	>1	
15	04.11	04.11		Juvenile	<1	
16	04.20	04.21	F	Adult		
17	04.20	04.21		Juvenile		
18	04.21	04.29				
19	04.21	04.21		Juvenile		
20	04.21	04.21		Juvenile		

## Data Availability

Data available on request due to restrictions.

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
