# Peer review of "Habitat Characteristics Coincidence of Dead and Living Long-Tailed Gorals (Naemorhedus caudatus) According to Extreme Snowfall"

_animals, 2021, doi:10.3390/ani11040997_

Round 1
Reviewer 1 Report
[General comments]
In this study the authors studied habitat characteristics of gorals, and tried to address differences in the characteristics between living and dead animals. Then authors found that gorals were dead due to climate catastrophe, and mortality was greater in pregnant females and infants. The research is well designed and analyses seems appropriate. And, authors' finding is useful for conservation act of endangered animals. Thus I would like to support this MS to be appeared in journal "Animals" after the revision.
I have one question...if "age" and "dominant rank" of animals are proportional, intraspecific competition (i.e., male > female and adult > juvenile) might have also affected the mortality. Do you have evidence that only heavy snow affected the mortality?
Followings are specific comments. I hope that you make use them to improve the MS.
--------------------------
[Specific comments]
Introduction
L44: "Naemorhedus" in Italic
L54: "automobiles" Does it mean road-kill? Please specify the issue.
L59: "abnormal heavy snow" You should show data (as a figure?) about abnormal weather in 2010.
L60-61: "have been" -> "were", "have died" -> "died"
L68: "gorals" -> "animals"
Materials and Methods
L86: "Fagaceae" is in non-Italic, or "Quercus (in Italic)."
L136: "species distribution models" Better to show abbreviation (SDM) here, because you used the abbreviation at L174.
Discussion
L241: "27" see my above comment.
L264-265: "could not rejected" -> "was rejected" or "was not supported"
L274: You need reference(s).
References
L359: Add "(" before species name.
L411: Article #55 is not cited.
Figures and Tables
*Information in figures was too small and I could not check well.
Table 1. I think that there are only 22 factors (not 27). Please check again. Regarding it, you may add # of factor (1-22) in the left side of the table.
Author Response
In this study the authors studied habitat characteristics of gorals, and tried to address differences in the characteristics between living and dead animals. Then authors found that gorals were dead due to climate catastrophe, and mortality was greater in pregnant females and infants. The research is well designed and analyses seems appropriate. And, authors' finding is useful for conservation act of endangered animals. Thus I would like to support this MS to be appeared in journal "Animals" after the revision.
I have one question...if "age" and "dominant rank" of animals are proportional, intraspecific competition (i.e., male > female and adult > juvenile) might have also affected the mortality. Do you have evidence that only heavy snow affected the mortality?
- We thank you for your assessment. I hope this manuscript will be helpful to make a conservation planning.
- Yes, we have the information published in the academic journal. The numbers of adults (5 ± 3, Average ± SE) were higher than those of juveniles (3 ± 2). Besides, the numbers of females (4.5 ± 0.5) were higher than those of males (2.5 ± 0.5). However, even though the sites were the representative habitats for the goral, the number of sites in the study area was only two. Thus, we wrote the possibility of your assumption as the study limitation (Lines 303–311).
Followings are specific comments. I hope that you make use them to improve the MS.
--------------------------
[Specific comments]
Introduction
L44: "Naemorhedus" in Italic
- Thank you for your advice (Line 45).
L54: "automobiles" Does it mean road-kill? Please specify the issue.
- We corrected it. Thanks (Line 56).
L59: "abnormal heavy snow" You should show data (as a figure?) about abnormal weather in 2010.
- We showed the data as a figure (Lines 64–67; Figure 1).
L60-61: "have been" -> "were", "have died" -> "died"
- We corrected it (Lines 62–63).
L68: "gorals" -> "animals"
- We corrected it (Line 79).
Materials and Methods
L86: "Fagaceae" is in non-Italic, or "Quercus (in Italic)."
- We corrected it (Line 96).
L136: "species distribution models" Better to show abbreviation (SDM) here, because you used the abbreviation at L174.
- We sincerely appreciated your careful comment. We corrected it (Line 148).
Discussion
L241: "27" see my above comment.
- We corrected it. As we addressed the name of variables, we did not put the details to miscalculate the number of the variables. Now, we described full names and numbered them (Table 1). We also added the units of the variables.
L264-265: "could not rejected" -> "was rejected" or "was not supported"
- Thank you for your comment. I corrected it (Lines 280–281).
L274: You need reference(s).
- We added some references (Line 291).
References
L359: Add "(" before species name.
- We added it (Line 395).
L411: Article #55 is not cited.
- We changed the number 55 to 57 (Line 449).
Figures and Tables
*Information in figures was too small and I could not check well.
- We enlarged all figures.
Table 1. I think that there are only 22 factors (not 27). Please check again. Regarding it, you may add # of factor (1-22) in the left side of the table.
- We thoroughly appreciated your valuable comments. We numbered all variables (Table 1).

Reviewer 2 Report
Overall I found this paper to be well written and informative, with mainly minor issues to resolve before it is acceptable for publication. The presentation of data and figures is generally of high quality. My primary concern is that the authors seem to reach firm conclusions that the data presented does not necessarily support. You focus almost exclusively on snow events as the cause of mortality in this population of animals - unless the deaths recorded have been confirmed to be the result of extreme weather (if so, please make this clear), it would improve the paper to at least show some consideration of alternative risk factors. Otherwise, I believe the paper is close to acceptable.
39 – Is “green” food the appropriate term here? There are nutrients in plant matter that would not necessarily be considered green (e.g. dried grasses), so I think perhaps use of the term food alone (or “plant matter”) may be more technically accurate.
51-52 – It may be valuable to explain how living in deep valleys and cliffs helps this population survive.
55 – You note that heavy snow “could” result in death losses; aside from in this paper, but then cite situations in which this occurred. Perhaps rephrase to something like “heavy snow represents a mortality risk.” Obviously minor issue up to authors’ discretion.
60 – Are these numbers accurate? Obviously, with several decades between these events, one can envision the population drop-off but the difference is huge relative to the total population.
67 – Change to “Gorals live in flexible, hierarchical…” or similar.
95 – How were the high and low population density areas defined? Was it simply by density as defined in the statistical models or were the regions predefined?
122 – There must me more detail about what is meant by considering male range – how was this measured?
148 – It may be helpful to list the variables that were removed.
238 – They may have been accurate based on the data considered, so may be better to say that they did not have a complete view.
271 – I’m not sure that this is sufficient to exclude social rank from factors influencing mortality. Even though mortality is highest in so called “preferred” habitats (which one would expect, as these habitats would have higher population density in general), access to resources within that habitat are still influenced by competition between animals as is seen in other ungulate species occupying the same territory.
274-286 – This information (and table 2) must be included somewhere in the results section. Was it confirmed that each of these animals died from starvation due to heavy snow? It seems that the authors do not consider other possible factors that might influence mortality (e.g. populations of goral living at higher density may be more at risk of transmitting disease between individuals). We know that the postnatal period has higher mortality in most ungulate species, regardless of extreme weather, so it is entirely possible that other risk factors may be relevant here.
Author Response
Overall I found this paper to be well written and informative, with mainly minor issues to resolve before it is acceptable for publication. The presentation of data and figures is generally of high quality. My primary concern is that the authors seem to reach firm conclusions that the data presented does not necessarily support. You focus almost exclusively on snow events as the cause of mortality in this population of animals - unless the deaths recorded have been confirmed to be the result of extreme weather (if so, please make this clear), it would improve the paper to at least show some consideration of alternative risk factors. Otherwise, I believe the paper is close to acceptable.
39 – Is “green” food the appropriate term here? There are nutrients in plant matter that would not necessarily be considered green (e.g. dried grasses), so I think perhaps use of the term food alone (or “plant matter”) may be more technically accurate.
- We corrected it to just “food” (Line 40).
51-52 – It may be valuable to explain how living in deep valleys and cliffs helps this population survive.
- We added the information (Lines 52–54).
55 – You note that heavy snow “could” result in death losses; aside from in this paper, but then cite situations in which this occurred. Perhaps rephrase to something like “heavy snow represents a mortality risk.” Obviously minor issue up to authors’ discretion.
- We corrected it (Line 57).
60 – Are these numbers accurate? Obviously, with several decades between these events, one can envision the population drop-off but the difference is huge relative to the total population.
- The number of individuals in the 1960s was much higher than those of individuals in recent years. The number is not accurate but estimated by a scientist.
67 – Change to “Gorals live in flexible, hierarchical…” or similar.
- We corrected it (Line 78).
95 – How were the high and low population density areas defined? Was it simply by density as defined in the statistical models or were the regions predefined?
- The regions were predefined by the small watershed, which could separate the regional habitats for the gorals. The densities between the two areas were different without any statistical models evidenced by the density of the traces such as feces, marking, etc. Because the lower areas have been recolonized near the survey period (Lines 102–105).
122 – There must me more detail about what is meant by considering male range – how was this measured?
- We added information on how the range was measured (Line 134).
148 – It may be helpful to list the variables that were removed.
- We listed the variables (Lines 162–164).
238 – They may have been accurate based on the data considered, so may be better to say that they did not have a complete view.
- We thank you for your comment. We corrected it (Lines 253–254).
271 – I’m not sure that this is sufficient to exclude social rank from factors influencing mortality. Even though mortality is highest in so called “preferred” habitats (which one would expect, as these habitats would have higher population density in general), access to resources within that habitat are still influenced by competition between animals as is seen in other ungulate species occupying the same territory.
- We thank you for the comment. We revised the sentence (Lines 287–288).
274–86 – This information (and table 2) must be included somewhere in the results section. Was it confirmed that each of these animals died from starvation due to heavy snow? It seems that the authors do not consider other possible factors that might influence mortality (e.g., populations of goral living at higher density may be more at risk of transmitting disease between individuals). We know that the postnatal period has higher mortality in most ungulate species, regardless of extreme weather, so it is entirely possible that other risk factors may be relevant here.
- We appreciated your comments. We could not give the information in the results section because we did not evolve to investigate the autopsy of the dead gorals. The report suggested that the gorals have significantly suffered from starvation, but any other disease symptoms were not shown. Thus, we concluded that mobility could be a problem to survive. We corrected and added this information. Please, see the revised manuscript (Lines 289–291).

Reviewer 3 Report
Interesting work which in a scientifically appropriate way analyzes a specific problem in the field of protection of an endangered species of mammal. Any possible reservations regarding, for example, the size of the research material result directly from the availability (dead animals from the endangered species). The authors do not propose specific technical solutions, but the work is not devoted to this, so it is not a fault. Possible linguistic errors may possibly be indicated by a reviewer who is a native English speaker. I rate the work highly and believe that it is worth publishing.
Author Response
Interesting work which in a scientifically appropriate way analyzes a specific problem in the field of protection of an endangered species of mammal. Any possible reservations regarding, for example, the size of the research material result directly from the availability (dead animals from the endangered species). The authors do not propose specific technical solutions, but the work is not devoted to this, so it is not a fault. Possible linguistic errors may possibly be indicated by a reviewer who is a native English speaker. I rate the work highly and believe that it is worth publishing.
- We appreciate your thorough comments. We have revised the manuscript according to the comments of the reviewers and improved English by the company. Please, see the revised manuscript.
